# Ion rectification based on gel polymer electrolyte ionic diode

Fan Jiang[1], Wei Church Poh[1], Juntong Chen[1], Dace Gao[1], Feng Jiang[1], Xiaoyu Guo[1], Jian Chen[1] & Pooi See Lee [1] ✉

Biological ion channels rely on ions as charge carriers and unidirectional ion flow to produce and transmit signals. To realize artificial biological inspired circuitry and seamless human-machine communication, ion-transport-based rectification devices should be developed. In this research, poly(methyl methacrylate) (PMMA) and poly(vinylidene fluoride-co-hexafluoropropylene) (PVDF-HFP) gel polymer electrolytes (GPEs) are assembled to construct a novel ionic diode, enabling ion rectification through ion-diffusion/migration that emulates biological systems. This ion rectification results from the different diffusion/migration behaviors of mobile ions transporting in the GPE hetero-junction. The electrical tests of the GPE heterojunction reveal outstanding rectifying ratio of 23.11. The GPE ionic diode operates in wide temperature window, from −20 °C (anti-freezing) to 125 °C (thermal tolerance). The absence of redox reactions is verified in the cyclic voltammogram. The GPE ionic diodes are used to construct ionic logic gates for signal communication. Furthermore, rectification of a triboelectric nanogenerator and potential for synaptic devices are demonstrated.

Biological systems rely on the transport of ions ($Na^+$, $K^+$, $Ca^{2+}$, $Cl^-$) and molecules to transmit information[1], operating in the form of ion channel/ion pump and producing nonlinear signals. SCN3A and SCN9A, for instance, are voltage-gated ion channels that are responsible for the $Na^+$ influx in sweet, bitter, and umami taste bud cells[2]. Similarly, ion-transport-based devices have emerged as a promising platform to control ionic charge carriers, emulating the information transmission process in biological systems[3]. The distribution and diffusion of ions in ion-transport-based devices could be controlled and manipulated by ion concentration and structural design, resulting in the generation, transmission and storage of signals[4]. Thus, different from electron-based solid-state electronics, ion-transport-based devices possess a great potential in achieving seamless communication with biological systems since they speak the "same language"[5]. In a biological ion channel, unidirectional ion transport is triggered by different stimuli, performing a diode-like rectifying behavior with ionic charge carriers[6]. This ion rectification realizes accurate modulation of ion transport in artificial devices[7].

Various structures and mechanisms have been reported to attain ion rectification. Typically, ion rectification happens when ions are either trapped or transported in an asymmetric structure (hetero-junction) to incur nonlinear current-voltage (I-V) curves[8–15]. This is unlike an electrolytic ionic diode which produces rectification by asymmetric faradaic reactions at electrodes, and thereby, essentially categorized as an electron-based rather than ion-transport-based system[16,17]. Ion rectifications performed through nanopore or nano-channel heterojunctions have been reported by Yan's and Karnik's groups, through electrostatic interactions from surface charges when the channel reached nanoscale[9,11]. Polyelectrolyte ionic diode (PID) stood out as solid-state ion rectification system[18,19]. The PID brings about the rectifying performance by generating an ionic double layer (IDL) to control ionic movement, analogous to a conventional p-n junction that controls electron transportation[20]. For example, Cayre et al. developed an agarose-based hydrogel polyelectrolyte heterojunction with a rectification ratio of 10 under ±5 V[21]. Han et al. developed a rectifying system with poly(diallyldimethylammonium

[1]School of Materials Science and Engineering, Nanyang Technological University 50 Nanyang Avenue, Singapore 639798, Singapore.
✉e-mail: pslee@ntu.edu.sg

chloride) (pDADMAC) and poly(2-acrylamido-2-methyl-1-propane-sulfonic acid) (pAMPSA) as the polyelectrolytic plug to execute ion selection[22]. The ionic diode fabricated with ionoelastomer achieved by Kim et al. has also drawn much attention, as their device is free from common problems caused by highly fluidic liquid electrolyte[23].

In biological systems, ions are evenly distributed inside or outside of cells at resting state, and then transport via electrochemical gradient[24]. Thus, at present, the typical ion rectification devices do not resemble the biological systems, as biological cells do not rely on fixed charged backbones or surfaces. In addition, the reported ion rectification devices still possess obvious deficiencies and issues. Firstly, many polyelectrolytes are hydrogels, thus these PIDs suffer from limitations related to the water electrolytes, such as evaporation and narrow electrochemical window (e.g. 1.23 V vs. Ag/Ag⁺)[25]. Secondly, ionoelastomer often exhibits poor ionic conductivities attributed to restrained transport of mobile counterions in the elastomer due to the absence of liquid media and electrostatic attraction by the charged polymer backbones[26]. Thirdly, nanopore/nanochannel devices require sophisticated processing and high costs to realize the charged surfaces.

In this work, in order to develop next-generation ion rectification system that can better mimic ion transport in the biological process, we design and fabricate an ion rectification ionic diode by using gel polymer electrolytes (GPEs), which leveraged on the novel mechanism of ion-diffusion-migration. In the GPE ionic diode, both positive and negative ions are mobile and freely diffuse without being attracted or repelled by any charged polymer or surface. The high or low diffusion/migration rates of ions in different GPEs are used to produce a preferential ion flow at the GPE interface, realizing the ion rectification. Judicious selection and preparation of GPEs with high-boiling-point, low volatility organic solvent and hydrophobic polymer matrices mitigate problems related to solvent evaporation and ensure high thermal stability. Combined with the electrochemically stable ionic liquid, salts, and organic solvent as the electrolyte, the absence of electrochemical redox reactions (or being non-faradaic) can be committed. To demonstrate the rectifying performance and versatile integration into other functional devices, we integrated the GPE ionic

diode with electronic resistors for logic gates construction for signal transmission and triboelectric nanogenerator (TENG) for energy harvesting. The potential of the ionic diode as a synaptic device is also displayed.

## Results and discussion

The GPE ionic diode is assembled by interfacing two disparate GPEs, constructing a heterojunction. The first GPE (denoted as PAZT), comprises of poly(methyl methacrylate) (PMMA) as the polymeric matrix, propylene carbonate (PC) as the solvent, and zinc triflate ($Zn(CF_3O_3S)_2$) as the ionic moiety. While for the second GPE (denoted as PHEC), poly(vinylidene fluoride-co-hexafluoropropylene) (PVDF-HFP)/PC system is selected, adopting 1-ethyl-3-methylimidazolium chloride ([EMIM]Cl) as the ion source. The PMMA/PC is chosen due to its good thermal stability and flexibility[27–29], and PVDF-HFP/PC is selected for its good plasticity and mechanical strength[30,31]. PC as the solvent for both matrices does not only serve to facilitate ion diffusion and migration with high ionic conductivities, but also promises thermal and electrochemical stabilities[32,33].

As depicted in Fig. 1a, $[EMIM]^+$ and $CF_3O_3S^-$ ions are mobile and well-dispersed in their respective GPEs. Since $Cl^-$ ions from PHEC exhibit limited ion diffusion in PAZT, the $Cl^-$ ions experience restricted ion transport when passing through GPE interface and entering PAZT. Likewise, $Zn^{2+}$ ions from PAZT are difficult to diffuse into PHEC for the same reason. Once the two GPEs are in contact, $Zn^{2+}$ and $Cl^-$ would diffuse and accumulate at the interface to form an IDL due to ion concentration gradient across the device, ceasing further diffusion of mobile $[EMIM]^+$ and $CF_3O_3S^-$ through the interface.

Under a forward bias (Fig. 1b), $[EMIM]^+$ ions in the PHEC and $CF_3O_3S^-$ ions in the PAZT GPE would be drawn towards the interface (Fig. 1b). Owing to the high migration rates of these ions, they are less obstructed in the GPE and could pass through the interface into the other GPE. In other words, the forward voltage bias eliminates the IDL constructed by $Zn^{2+}$ and $Cl^-$ and assists the ions to flow through the device, leading to a relatively high current in the device and forward conduction. When a reverse bias is applied on the GPE ionic diode (Fig. 1c), the IDL would be enhanced by greater $Zn^{2+}/Cl^-$ interfacial

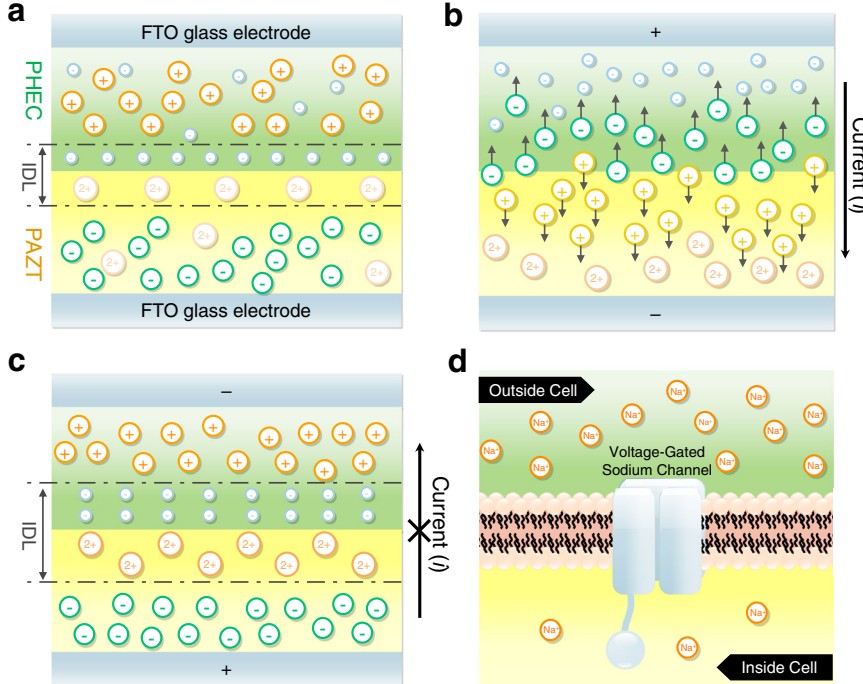

**Fig. 1 | Working mechanism of GPE ionic diode. a** schematic of GPE ionic diode without voltage bias; **b** under forward bias; **c** under reverse bias; **d** schematic of biological neuron ion channel.

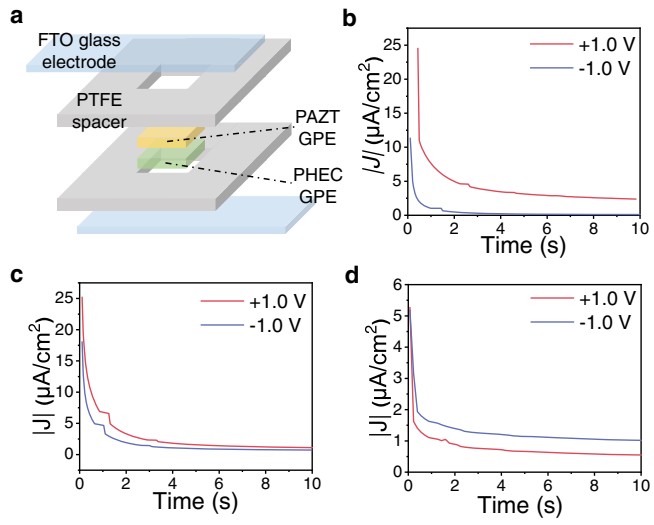

**Fig. 2 | Structure and transient current responses of GPE ionic diode. a** GPE ionic diode in sandwich structure; **b** transient current response of PAZT/PHEC heterojunction at ±1.0 V; **c** PHEC/PHEC homojunction at ±1.0 V; **d** PAZT/PAZT homojunction at ±1.0 V.

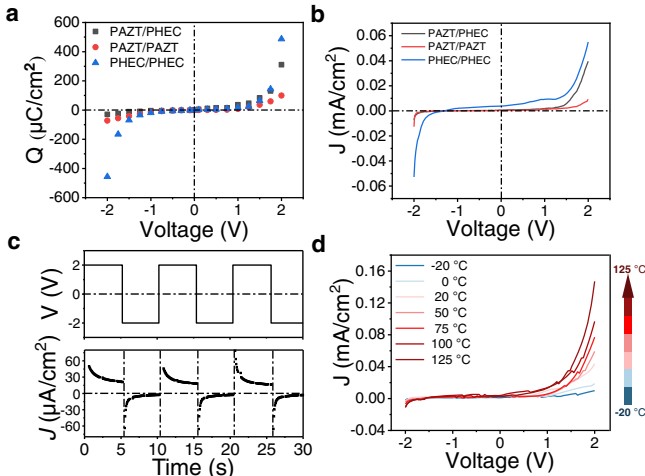

**Fig. 3 | Rectification by GPE ionic diodes. a** $Q$-$V$ plots of GPE homojunctions and heterojunction diode from −2 V to + 2 V; **b** directly measured $I$−$V$ curves for PHEC/PHEC, PAZT/PAZT and PHEC/PAZT; **c** rectification by the GPE ionic diode at alternating voltage of ± 2 V at 0.1 Hz; **d** $I$–$V$ curves of the GPE ionic diode under different temperatures.

accumulation, thereby, further restraining ion transport of [EMIM]$^+$ and CF$_3$O$_3$S$^-$. This thicker IDL behaves like a widened depletion region in the conventional p-n junction with a stronger field. The ion flow is inhibited by the interface, performing a reverse cut-off that could be observed in common Si-based diodes.

The working principle of the GPE ionic diode lies in the difference in ion diffusion/solubility in the GPE heterojunction, while biological ion channel (Fig. 1d) relies on chemical activation and specific ion recognition. When compared with ion channel, the GPE ionic diode shares certain similarities in their approach of controlling ion transport. The ion rectification of GPE ionic diode is realized via permitting or restricting the flowing of certain ions through the GPE heterojunction, comparable in certain respects of ion-selective permeability in biological ion channels. In addition, unlike typical ion rectification devices which have fixed charges on polymer chains or surfaces, GPE ionic diode also shares common feature with biological system, that is, they both allow free ion transport of positive and negative ions.

The GPE ionic diode was assembled by layering PHEC and PAZT GPEs in a customized polytetrafluoroethylene (PTFE) mold (Fig. 2a). The PTFE spacers not only shape the GPE into a desired dimension but also protect the GPE ionic diode from direct environmental exposure. T-peel test (Supplementary Fig. 1) of GPE heterojunction revealed cohesive failure occurred within PAZT GPE, indicating strong and durable adhesion at PAZT/PHEC interface. To connect the fabricated GPE ionic diode to the external circuit for electrical connection, the PAZT/PHEC heterojunction was sandwiched between two fluorine-doped tin oxide (FTO) glass electrodes.

To characterize the ionic rectifying performance, we applied direct current (d.c.) voltages in both polarities to the PAZT/PAZT, PHEC/PHEC homojunctions and PAZT/PHEC heterojunction. For the heterojunction PAZT/PHEC (Fig. 2b), ±1.0 V bias produces currents of 2.38 and − 0.10 μA/cm² respectively, demonstrating rectification ratio ($\eta$) up to 23.11 at stabilized stage. The forward voltage bias eliminates the IDL and assists [EMIM]$^+$ and CF$_3$O$_3$S$^-$ of high ion migration to flow through the device, leading to relatively high current in the device; while reverse bias reinforces IDL by drawing more Cl$^-$ and Zn$^{2+}$ towards GPE interface, inhibiting ion flow through the ionic diode. In contrast, the current density within the PHEC/PHEC and PAZT/PAZT homojunctions under positive and negative voltage are comparable (no rectification) even though PHEC/PHEC produces higher current density than the PAZT/PAZT due to the higher ionic conductivity (Fig. 2c, d).

The transient current responses of PAZT/PAZT and PHEC/PHEC homojunctions under ± 1.0 V, ± 1.5 V and ± 2.0 V are shown in Supplementary Fig. 2a–i and summarized in Supplementary Table 1. Under a prolonged period of ± 1.0 V in Supplementary Fig. 3a, the rectifying ratio reached high value of 24.0 at 3.5 s, then decayed to 80% ($\eta$ = 19.2) at 29.5 s and 50% ($\eta$ = 12.0) at 114.4 s.

The $Q$-$V$ curves of the two homojunctions and heterojunction are presented in Fig. 3a and used to derive the total associated charges by integrating the area under the transient current curves within the stable voltage bias range (−2.0 V to +2.0 V). Obvious symmetry could be observed in both PAZT/PAZT and PHEC/PHEC homojunctions. Unlike homojunctions, clear asymmetry could be seen in the $Q$-$V$ curve of the PAZT/PHEC heterojunction, which rises exponentially at around 0.7 V. Based on the differences of $I$-$t$ and $Q$-$V$ curves between heterojunction and homojunction, we could verify that the PAZT/PHEC heterojunction can act as an ionic diode with rectifying effect, with a turn-on voltage of ~0.7 V. The current−voltage ($I$−$V$) characteristics were directly measured with the voltage sweeping from a reverse bias (−2.0 V) to a forward bias (+ 2.0 V). Figure 3b shows the $I$−$V$ curves of homojunctions and heterojunction diode. Under + 2.0 V forward bias, the current passing through PAZT/PHEC heterojunction reaches 0.039 mA/cm², which is 6.72 times larger than the current under the reverse bias (−0.0058 mA/cm²). Compared with symmetric $I$−$V$ curves of the homojunctions PAZT/PAZT and PHEC/PHEC, the ionic diode possesses obvious rectifying capability resulting from the IDL.

The rectifying ability of the ionic diode was further examined under an alternating potential. As illustrated in Fig. 3c, the PAZT/PHEC heterojunction displays obvious rectification under cycles of alternating voltage at ±2 V (0.1 Hz). In Supplementary Fig. 3b, under a square-wave voltage of ±1 V at 0.1 Hz, the device could preserve 75.8% of its initial rectification ratio after 50 cycles. To evaluate the extreme temperature tolerance of GPE ionic diode, $I$−$V$ curves (−2.0 V to +2.0 V) of GPE ionic diode under different temperatures (−20 °C to 125 °C) were measured and the tested results are displayed in Fig. 3d. It was found that the GPE ionic diode performed good ion rectification under different temperatures. The rectifying ratios are 11.59 at 20 °C and reach 13.02 when the temperature gradually increased to 125 °C. The increased current density is a result of faster ion movement at the elevated temperature. In contrast, the rectifying ratio decreases to ca. 5 at −20 °C, which could be caused by restrained ion dissociation of salt and ionic liquid under freezing temperature. The device is able to

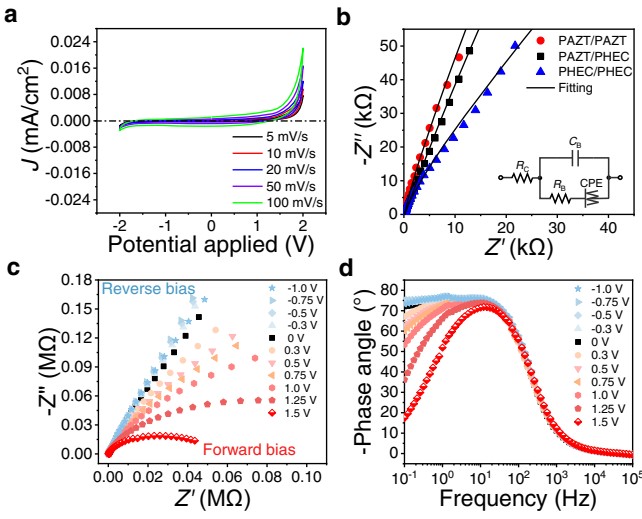

**Fig. 4 | Electrochemical analysis of GPE ionic diode. a** CV plots for PAZT/PHEC scanned from −2 V to + 2 V at various scan rates; **b** Nyquist plots for PHEC/PHEC, PAZT/PAZT and PHEC/PAZT; **c** Nyquist plot of PAZT/PHEC heterojunction measured under dc biases; **d** Bode phase plot of PAZT/PHEC heterojunction measured under dc biases.

maintain stable electrical performance under extreme temperatures owing to the use of high boiling point of PC and thermally stable solvated salts.

As illustrated in the cyclic voltammograms (CV) of GPE ionic diode in Fig. 4a and homojunction in Supplementary Fig. 6, no redox reaction peak or electrochemical reaction could be observed, owing to good stability and wide electrochemical window of PC solvent and ionic liquids. In addition, the magnitudes of current density obtained in all CV curves are well below the cut-off limit of 0.1 mA/cm² selected to identify the electrochemical window[34]. Therefore, this can verify that the GPE ionic diode is stable within the working range from − 2.0 V to 2.0 V without involving any electrochemical reaction (non-faradaic) or obvious electrolysis. In this regard, the high rectifying ratio of our device can be solely attributed to the current signals produced by ion diffusion and migration, without participation of a redox process. To further verify this, X-ray photoelectron spectroscopy (XPS) has been employed to chemically characterize the GPE ionic diode after applying voltages at − 2 V and 2 V for 10 s. As shown in Supplementary Figs. 4 and 5, the XPS analysis of PAZT and PHEC reveals that all the characteristic peaks correspond well with the ionic liquid, salt and polymer employed, without a trace of redox byproducts, which is in line with the CV results.

To provide insights into the Nyquist plots (Fig. 4b) and Bode phase plots (Supplementary Fig. 7) of homojunctions and heterojunction, an equivalent circuit model was used to generate fitting parameters which are summarized in Supplementary Table 2. The derived ionic conductivity of the PAZT/PAZT GPE is 1.2 ± 0.1 mS/cm and PHEC/PHEC is 1.6 ± 0.2 mS/cm. These high ionic conductivities are a result of enhanced ion mobility and dissociation provided by the PC plasticizer, which also reduces the crystallinity of polymer matrices for better migration conditions[35]. For the heterojunction PAZT/PHEC, the capacitance at the interface is 1.4 ± 0.3 µF/cm², indicative of the presence of an IDL at the heterojunction interface. In contrast, PAZT/PAZT and PHEC/PHEC homojunctions reveal lower capacitance values (0.3 ± 0.1 and 0.2 ± 0.1 µF/cm²), as there are interdiffusion of ions across the junction, resulting in lower ions accumulation at the interface. The SEM-EDS elemental analysis (Supplementary Fig. 8) of GPE ionic diode also matches well with the IDL-based deduction. As shown in Supplementary Fig. 8 and Supplementary Table 3, Cl element can be detected at PHEC near the interface while N element (from EMIM⁺ ions)

cannot be found, and the measured S/Zn ratio of 0.67 at PAZT near the interface is much lower than that of bulk PAZT (2.13). This result indicates accumulations of Cl⁻ and Zn²⁺ ions near the PAZT/PHEC interface, while lower amount of [EMIM]⁺ and CF₃O₃S⁻ exist near the interface.

To further analyze the IDL of PAZT/PHEC heterojunction, a series of EIS was measured upon application of different DC biases. As shown in Fig. 4c, an obvious decrease in impedance could be observed in the low-frequency region of the Nyquist plot when the forward bias is gradually increased. In addition, the phase angle in the Bode phase plot (Fig. 4d) also rapidly declines in the low-frequency region when the applied direct voltage stepped up from 0 V to 1.5 V. The decreased impedance in Fig. 4c can be attributed to $R_B$ which falls in correspondence with faster ion migration under stronger forward bias, allowing ions to pass through the interface more easily. In addition, the weakened IDL under a forward bias also leads to the reduction of IDL capacitance, which is suggested by the decreased phase angle in the low-frequency zone. On the contrary, a slight increase in impedance and phase angle could be observed under a reverse bias. This phenomenon indicates the existence of thicker IDL with poorer ionic migration, behaving like a widened depletion region in the conventional p-n junction.

To demonstrate the real application of the GPE ionic diode, a triboelectric nanogenerator (TENG) was deployed to test the ion rectification of the GPE ionic diode. A TENG which produces periodic positive and negative voltages (AC characteristic), requires a rectifier connected in series to afford DC for energy storage. A TENG circuit with an ionic diode and an energy storage device was set up as shown in Fig. 5a. As demonstrated in Fig. 5b, after 5 consecutive rubbing on the TENG, the voltage of capacitor increases from −0.74 mV to 70.46 mV. During the relaxation period of 5 s the voltage charged in the capacitor dissipates and decreases. Over 6 repeated charging-resting cycles, the capacitor gains overall voltage charging to 97.01 mV. When the GPE ionic diode is reversed after the 6th cycle, the capacitor charges negatively by the TENG and displays reverse charging from 96.30 mV to −18.32 mV after 6 cycles of rubbing and relaxation. For comparison, the capacitor circuit without GPE ionic diode only shows a small voltage fluctuation that is <0.85 mV (Fig. 5b) with no rectification under the same rubbing pattern of TENG. Thus, the ion rectification of the GPE ionic diode upon TENG can be verified.

To verify the potential of the GPE ionic diode in biological circuitry and human-machine interface, basic "AND" and "OR" logic gates were built. As derived from the previously measured I-V curves, we could determine the turn-on voltage of the ionic diodes at around 0.7 V. When the applied voltage bias exceeds the 0.7 V threshold, the ionic diode becomes conductive, or otherwise. The circuitry of OR gate based on the GPE ionic diode is shown in Fig. 5c. When voltage inputs A and B are higher than the turn-on voltage of the ionic diode, the input signal could be considered as "1"; the input signal would be "0" when there is no voltage input. Likewise, the high voltage measured at output C is regarded as output signal "1", and low output voltage would be output signal "0". To assess the designed ionic diode, voltage signals with 4 logic states (0,0), (0,1), (1,0), (1,1) were generated by the square waves from A and B. The measured output $V_C$ is shown in Fig. 5d. When the input voltage of either A or B is 1.5 V (logical "1"), the output voltage $V_C$ is high or in the logical state "1" (the red dash line around 0.9 V). When both $V_A$ and $V_B$ are 0 V (logical "0"), output $V_C$ is low or in logical "0". Based on these outcomes, the logical expression: C = A OR B in the ionic OR gate could be verified. Similarly, an ionic AND gate based on two GPE ionic diodes is also designed as demonstrated in Fig. 5e. As shown in Fig. 5f, in the case that either input signal $V_A$ or $V_B$ is 0 V (logical "0"), the two ionic diodes would be turned on by the $V_{CC}$ (2 V) and conduct current, leading to a low output $V_C$ of 0.6 V or logical "0" (under black dash line). However, when both $V_A$ and $V_B$ are of high voltage 1.5 V, the output $V_C$ is high or in logical "1" (above red dash line

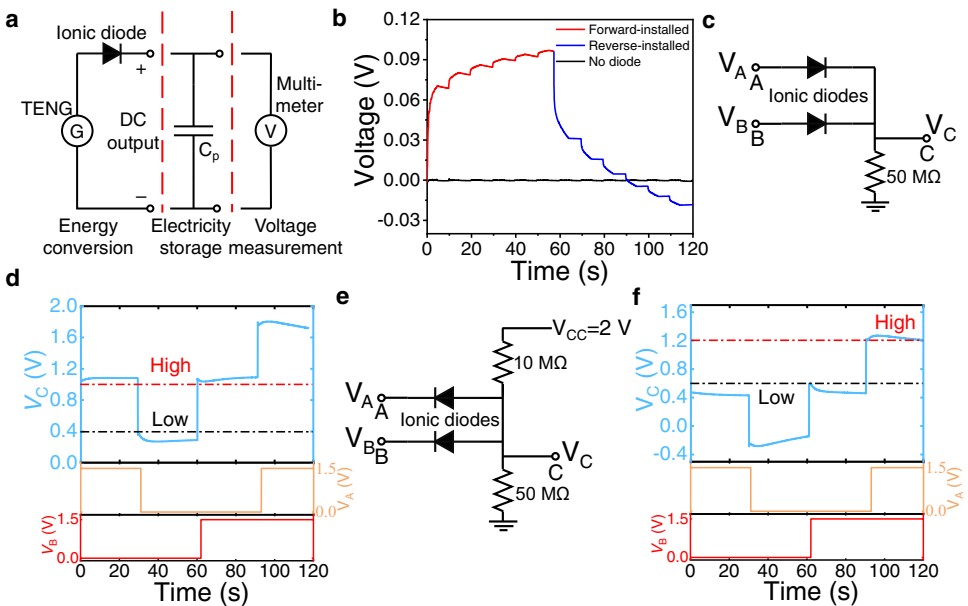

**Fig. 5 | Rectification of TENG and ionic logic gates by GPE ionic diodes. a** a rectifying circuit to converting AC produced by TENG; **b** capacitor voltage of the TENG as a function of time; **c** A circuit diagram of an ionic OR gate. A and B are the input terminals and C is the output terminal; **d** performance of ionic OR gate. Square waves with are programmed as binary inputs A and B for the OR gate.

Voltage at C is measured as the output; **e** a circuit diagram of an ionic AND gate. A and B are the input terminals and C is the output terminal; **f** performance of ionic AND gate. Square waves with are programmed as binary inputs A and B for the AND gate. Voltage at C is measured as the output.

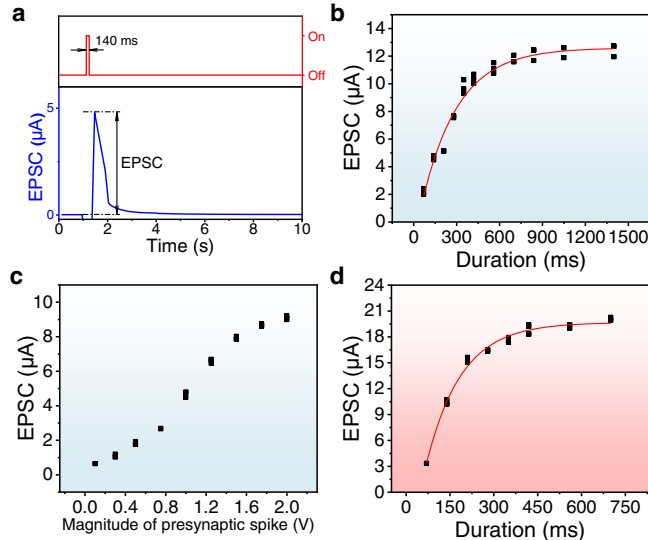

**Fig. 6 | EPSC of GPE ionic diodes. a** EPSC induced by a −1 V presynaptic voltage with the duration of 140 ms; **b** EPSC as a function of spike duration induced by the −1 V presynaptic voltage; **c** EPSC as a function of presynaptic voltage with spike width maintained at 140 ms; **d** EPSC as a function of spike duration induced by the −1 V presynaptic voltage under high temperature of 100 °C.

around 1.2 V). Therefore, we demonstrated the logical expression: C = A AND B in the ionic AND gate.

In order to explore the potential of GPE ionic diode in synaptic devices (neuromorphic properties), a sequence of short voltage pulses was applied and the corresponding responses of GPE ionic diode were analyzed. Figure 6a shows a typical excitatory postsynaptic current (EPSC) induced by a presynaptic voltage pulse of − 1 V with a duration of 140 ms. The current increases with the voltage spike and reaches the peak at 4.79 μA, then decays gradually to the baseline. As shown in Fig. 6b, the produced EPSC increases when the duration of presynaptic voltage (or the pulse width) is extended. The EPSC starts from

2.19 ± 0.16 μA at a pulse width of 70 ms and is saturated at a pulse width of 1400 ms, reaching a maximum of 12.47 ± 0.35 μA. This phenomenon is actuated by ion redistribution when the voltage pulse breaks the equilibrium in the GPE heterojunction, which resembles an EPSC in a biological synapse. The responses to incremental pulse amplitude from 0.1 V to 2.0 V with a constant pulse width of 140 ms are also investigated and shown in Fig. 6c. Due to the demand for intelligential applications in aerospace, deep-well drilling, and high-speed automobiles[36], EPSC is also demonstrated at an elevated temperature (Fig. 6d). Under a high temperature of 100 °C, the EPSC starts from 3.35 ± 0.02 μA at 70 ms pulse and plateaus at 20.07 ± 0.15 μA at 700 ms pulse. The premature saturation and higher EPSC is attributed to the accelerated ion diffusion and migration under high temperature environment.

In conclusion, we prepared GPEs and assembled rectifying ionic diode through an easy fabrication process for polymer ionic diode for ion modulation. One remarkable feature of this GPE ionic diode is that it operates on the basis of non-faradaic ion diffusion and migration, which mimics biological process. This property has been verified by electrochemical analysis, including EIS and CV. The GPE ionic diode possesses much higher temperature tolerance and thermal stability than water-based ionic diodes. The GPE integration of GPE ionic diode with TENG have been demonstrated. This GPE ionic diode also possesses unique features that has high potential for synaptic devices.

## Methods
### Material
Poly(vinylidene fluoride-co-hexafluoropropylene) (PVDF-HFP) pellets (Mw: 400,000, Sigma-Aldrich), poly(methacrylic acid methyl ester) (PMMA) powder (Mw: 996,000, Sigma-Aldrich), $Zn(CF_3O_3S)_2$ powder (98.0%, Sigma-Aldrich), 1-Ethyl-3-methylimidazolium chloride ([EMIM]Cl) (98.0%, Sigma-Aldrich), $ZnCl_2$ (≥98%, Sigma), $[EMIM]CF_3O_3S$ (≥ 98%, Sigma-Aldrich), [EMIM][TFSI] (≥98%, Sigma-Aldrich), Li[TFSI] (99.95%, Sigma-Aldrich), acetone (≥99.8%, Fisher Chemical), acetonitrile (≥99.9%, Fisher Chemical), propylene carbonate (PC) (99.7%, Sigma-Aldrich), fluorine-doped tin oxide (FTO) glass, and polytetrafluoroethylene (PTFE) spacers were purchased and directly used without further treatments.

## Preparation of GPE precursor solutions

The precursor solutions of GPEs were prepared through simple mixing and stirring process. For the PVDF-HFP/[EMIM]Cl (PHEC) GPE, 7.0 g acetone and 3.0 g PC were mixed through magnetic stirring. Then 1.0 g PVDF-HFP was added in the acetone/PC solvent and stirred at 50 °C for 1 hour. Finally, 0.2 g [EMIM]Cl was added in the PVDF-HFP/acetone/PC solution and stirred in ambient temperature for 1 hour to obtain a clear and homogenous precursor solution. For the PMMA/zinc triflate (PAZT) GPE, 1.0 g PMMA powder and 0.5 g $Zn(CF_3O_3S)_2$ were added in the mixture of 8.0 g acetonitrile and 3.0 g PC. After stirring at 60 °C for 1 h, transparent precursor solution for PAZT GPE was prepared.

## Solubility test

In order to satisfy the working mechanism of ionic diode, screening of cations and anions has been conducted through a series of salt solubility tests. According to the results of solubility tests (Supplementary Table 4), $Zn(CF_3O_3S)_2$/PVDF-HFP/PC gel experiences obvious and rapid phase separation, while bulk precipitation is observed in the chloride salt/PMMA/PC solution. Thus, as shown in Table S5, $[EMIM]^+$ and $CF_3O_3S^-$ ions could dissolve well in both PVDF-HFP/PC and PMMA/PC gel, leading to high current output under forward bias; while $Zn^{2+}$ ions are more difficult to dissolve in the PVDF-HFP matrix, and $Cl^-$ ions similarly could not dissolve well in PMMA matrix. In addition, through comparison of ion diffusion coefficients (Supplementary Fig. 9 and Supplementary Table 6), the added cations and anions were selected accordingly.

## Fabrication and characterization of ionic diode

A sandwich structure was adopted to build the diode, with the aim to realize higher rectifying performance by increasing the interface area of the heterojunction. In addition, enclosed PTFE spacers were adopted to protect the samples from the moisture, contamination and evaporation from environment. Chemically inert FTO glass (5 cm × 1.5 cm) was used as electrodes that connect ionic diode to external circuit as it can better resist the corrosion by the electrolyte. The precursor solutions were directly drop-casted on the FTO glass substrate and partially dried to form conductive GPEs in solid state. We firstly prepared PAZT by drop-casting the precursor solution in the spacer (1 cm × 1 cm × 0.2 cm) built with PTFE, followed by drying under ambient temperature for 45 minutes to evaporate acetonitrile. Then PHEC precursor solution was drop-casted in another PTFE spacer with FTO glass substrate and dried under ambient temperature for 30 minutes. Then the two spacers were brought together to assemble the ionic diode in a sandwich structure. The FTO glass would extend outside to connect the ionic diode to external circuits. The circuit for rectification upon TENG included one energy conversion and rectification part consisting of TENG and GPE ionic diode, one electricity storage part of a 33 μF capacitor, and one multimeter part for measuring the voltage stored in the capacitor. The energy is generated from rubbing the TENG with frequency of 1 Hz, which goes through the GPE ionic diode and gets converted into direct voltage to charge capacitor.

To evaluate the rectifying performances of the ionic diode, Keithley 2400 SourceMeter was used for measuring $I$-$V$ characteristics, transient current response, cycles under alternating current, and EPSC. The electrochemical properties of the device have been analyzed using Autolab PGSTAT128N electrochemical workstation to perform CV and EIS. CV measurements were conducted with various scan rates from 5 to 100 mV/s, and EIS was measured from $10^5$ to $10^{-1}$ Hz with an amplitude of 10 mV. Voltage output was measured by Keithley DAQ6510 Multimeter System. To measure the electrical performance at different temperatures, the GPE ionic diode was kept inside a vacuum oven (France Etuves XFL020) and refrigerator. The DC voltage employed in the logic gates was supplied by MCH-K3206D DC power supply. XPS was conducted with XPS Kratos AXIS Supra. Scanning Electron Microscopy (SEM) was measured through FESEM 7600F.

T-peel test was conducted with MTS Criterion Model 42 electromechanical universal test system, according to the procedures in the ASTM D1876-08 standard with a crosshead speed of 50 mm/min.

## Data availability

The source data that support the findings of this study are available in figshare with the identifier https://doi.org/10.6084/m9.figshare.21087853.

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

## Acknowledgements

The project is funded by the Ministry of Education Tier 1 Grant Award No. RT15/20.

## Author contributions

Fan J. implemented the project. Fan J., W.C.P., and Juntong C. designed and performed the experiments. D.G. contributed to gel preparation and mechanical testing. Feng J. and Jian C. carried out preparation and measurement of TENG and construction of logic circuits. X.G. performed electrochemical test and analysis of results. Fan J., W.C.P., and P.S.L. discussed results and wrote the manuscript. P.S.L. conceived the idea and supervised the entire project.

## Competing interests

The authors declare no competing interests.
