## [Peer Review File · Nature Communications]

REVIEWER COMMENTS

Reviewer #1 (Remarks to the Author):

- What are the noteworthy results? Yes, see comments

- Will the work be of significance to the field or other fields? how does it compare to literature?

Yes, it has novelty versus existing literatures

- Does the work support the conclusions/claims? Yes or is additional evidence needed?

- Are the flaws in data analysis, interpretation, and conclusion

Yes, the data analyses and findings are well supported. Yes, pls see Q1 and Q3 for comments

- Is the methodology sound? does the work meet the expected standards in your field? Yes

- Are there enough detail provided for the work to be reproduced? Yes

Comments

The present work offers an alternative method to construct ionic diode using controlled ionic mobility in different polyelectrolyte media. It is scientifically novel since it reveals a new mechanism for ionic current rectification, other than using crosslinked polyelectrolytes or Faradic reactions at electrodes. The fully mobile positive and negative ions in this system are notably emulating biological neurons, which may lay a foundation for the construction of artificial neurons. Besides, I think the authors have provided sufficient evidence, including EIS, EDS, and XPS, to support the proposed mechanism. However, I have certain questions with the presented data as listed below:

1. The author mentioned that the turn-on voltage of the GPE diode is 1 V (Fig. 3a). Usually the rectification ratio of the diode should be characterized above the turn-on voltage, however in Fig. 2b the author presented diode rectification at +1 V and -1 V instead of using higher voltage, which is

contrary to the common practices. A calculation of rectification ratio at ± 1.5 V seems more reasonable to me.

2. How is the stability of the ionic rectification? How long can the high rectification ratio in Fig. 1b last?

3. I think there's some inconsistency between the data in Fig. 2b and 3b. In Fig. 2b the quasi-stable current under +1 V is $\sim 3 \mu\text{A}/\text{cm}^2$, yet the current density measured in Fig. 3b under +1 V, during continuous voltage ramping, the current density is below $3 \mu\text{A}/\text{cm}^2$. Can the author provide some explanation to this- Is this due to different testing mode or inconsistency between devices?

4. I'm interested in the mechanical property at the PAZT/PHEC interface. Is this interface robust and durable? What's the interfacial toughness?

5. For the logic gates constructed using ionic diodes (both OR and AND), why the measured voltage under (0,0) state is not 0 but 0.4 V or -0.4 V?

Reviewer #2 (Remarks to the Author):

The authors have reported an ionic liquid based ionic diode by using two gel polymer electrolytes. They suggested the new mechanism of ionic diode by using the difference in ion solubility in two gels. By using the ionic liquid, they secured the thermal stability than ordinary hydrogel ionic diode. However, there are some unclear factors and major points that this reviewer would like to ask. Please consult the followings:

1. The authors said the ionic diode is bioinspired, but it is difficult to say that there are similarities between the neuron ion channel and the ionic diode as the working principle is completely different. The neuron ion channel uses ion selective membranes, but the ionic diode uses the ion solubility difference between two gel polymer electrolytes. In terms of functionality, the neuron ion channel only moves Na^+ ions in one direction, whereas ionic diode moves anions in the opposite direction along with movement of cations. It is hard to say that gel polymer electrolyte ionic diode is bioinspired without further explanation.

2. In figure 3 & figure 4, there are many experiments about performance of the ionic diode. However, experiments about durability of the ionic diode are missing. Additional experiments are

required to confirm the durability of the ionic diode(Cycle test, maximum operating voltage range test, maximum operating time test, etc.).

3. Mechanism of the ionic diode is only possible with certain combinations of ions. Are there any other ions & gels combinations that the device can work with?

Reviewer #3 (Remarks to the Author):

I looked through the article and suggest major revisions based on the comments below.

The novelty of this work is an ionic diode that is not based on water or polyelectrolytes and can work at High temperatures.

The authors need to make a better comparison between different ionic diode systems in this work to show where this system is better in terms of quantitative metrics and possibly also areas of use. The comparison between this Ionic diode and the bio channels is in my opinion not very instructive and indeed even misleading: After all this system is not water based whereas biology is and so are already reported ion rectifiers. Second, the ion channels act based on a completely different mechanism where chemical energy and specific ionic recognition shuffle ions.

I suggest to remove the word bioinspired from the title and also rewrite the parts that make the direct comparison between this system and biological ion channels.

There are many grammatical errors for example biological system relies (should be rely), is selected for their (should be its), are difficult to travel (should have lower diffusion rate or similar). This language must be reworked throughout.

Another problem is that the application of Triboelectric generator simply does not make sense to do with this system. The generator is purely electrical, and a regular solid state diode has much better properties for this application. The authors say that this diode is suitable for neuromorphic devices

in conclusion but have no application or data to verify this statement and ideally also show a simple application where they use the device for analog Computation.

In my opinion, if this article is to be published here there should at minimum be an application that is clearly relevant for this particular device, for example something which relies on high temperature above 100°C where this device is superior, and/or a system where the input to the diode is itself based on iontronic systems (eg an ionic sensor connected to diodes for computation or something like that)

Responses to Reviewer Comments for

Ion Rectification Based on Gel Polymer Electrolyte Ionic Diode

REVIEWER COMMENTS

Reviewer #1 (Remarks to the Author):

- What are the noteworthy results? Yes, see comments

- Will the work be of significance to the field or other fields? how does it compare to literature?

Yes, it has novelty versus existing literatures

- Does the work support the conclusions/claims? Yes or is additional evidence needed?

- Are the flaws in data analysis, interpretation, and conclusion

Yes, the data analyses and findings are well supported. Yes, pls see Q1 and Q3 for comments

- Is the methodology sound? does the work meet the expected standards in your field?

Yes

- Are there enough detail provided for the work to be reproduced? Yes

Comments

The present work offers an alternative method to construct ionic diode using controlled ionic mobility in different polyelectrolyte media. It is scientifically novel since it reveals a new mechanism for ionic current rectification, other than using crosslinked

polyelectrolytes or Faradic reactions at electrodes. The fully mobile positive and negative ions in this system are notably emulating biological neurons, which may lay a foundation for the construction of artificial neurons. Besides, I think the authors have provided sufficient evidence, including EIS, EDS, and XPS, to support the proposed mechanism. However, I have certain questions with the presented data as listed below:

1. The author mentioned that the turn-on voltage of the GPE diode is 1 V (Fig. 3a). Usually the rectification ratio of the diode should be characterized above the turn-on voltage, however in Fig. 2b the author presented diode rectification at +1 V and -1 V instead of using higher voltage, which is contrary to the common practices. A calculation of rectification ratio at ± 1.5 V seems more reasonable to me.

The turn-on voltage is defined as the point where current-voltage curve switches from the initial linear rising to an exponential rising. As shown in Fig. R1, the I-V curve first rises linearly from 0 to 0.7 V, then rises exponentially from 0.7 V and beyond. By doing linear fittings, the derived major R-squared value from 0 to 0.7 V is 0.983, while the R-squared value from 0 to 1.0 V reduces to 0.951, illustrating deviation from linearity starting after 0.7 V. Thus, based on a more careful analysis, the turn-on voltage of GPE ionic diode is redefined to be ~ 0.7 V here. Therefore, the rectifying ratio remains to be 23.11, which is characterized at ± 1.0 V above the turn-on voltage (~ 0.7 V)

Fig. R1 Analyzed I-V curves for PHEC/PAZT heterojunction.

2. How is the stability of the ionic rectification? How long can the high rectification ratio in Fig. 1b last?

To test the stability of ion rectification in GPE ionic diode, a prolonged period of ± 1.0 V is applied. As shown in Fig. S3(a), the current-time curves produced by the ± 1.0 V remains to be smooth and stable in the 200 s measurement, indicating good durability for long-term operation. The rectifying ratio (η) reaches high value of 24.0 at 3.5 s and starts to decrease at 12.6 s. The rectifying ratio decays to 80% ($\eta = 19.2$) at 29.5 s and 50% ($\eta = 12.0$) at 114.4 s, finally 41.4% ($\eta = 9.95$) at the 200 s. Moreover, the durability of the GPE ionic diode under a square-wave voltage of ± 1 V with 0.1 Hz frequency was also measured and displayed in Fig. S3(b). The rectifying ratio of GPE ionic diode decreased to 75.8% of initial performance after 50 cycles, then 56.2 % of initial performance remains after 100 cycles of alternating voltage.

Fig. S3 (a) Transient current response of PAZT/PHEC heterojunction at ± 1.0 V for 200 s; (b) performance stability of the PAZT/PHEC diode a square-wave voltage of ± 1.0 V with 0.1 Hz frequency.

3. I think there's some inconsistency between the data in Fig. 2b and 3b. In Fig. 2b the quasi-stable current under +1 V is $\sim 3 \mu\text{A}/\text{cm}^2$, yet the current density measured in Fig. 3b under +1 V, during continuous voltage ramping, the current density is below $3 \mu\text{A}/\text{cm}^2$. Can the author provide some explanation to this- Is this due to different testing mode or inconsistency between devices?

The current density in Fig. 2(b) is $2.38 \mu\text{A}/\text{cm}^2$ and the current density in Fig. 3(b) is $1.84 \mu\text{A}/\text{cm}^2$. The small discrepancy can be caused both by the difference in the two measurement processes (I-V and I-t) as well as minor batch-to-batch device variations. The results of the measured current density at +1 V with different prepared GPE ionic diodes are summarized in Table R1. The mean values of measured current densities illustrate small inconsistencies which are still considered acceptable.

Table R1. Measured current densities of GPE ionic diodes under +1.0 V with different testing methods.

Measurement type		Current density ($\mu\text{A}/\text{cm}^2$)	Mean
Transient current response (I-t)	1	2.38	2.31 ± 0.25
	2	2.58	
	3	1.98	

Continuous voltage ramping (I-V)	4	1.84	2.14 ± 0.31
	5	2.56	
	6	2.01	

4. I'm interested in the mechanical property at the PAZT/PHEC interface. Is this interface robust and durable? What's the interfacial toughness?

T-peel test was conducted to investigate the interfacial toughness of PAZT/PHEC heterojunction by using an MTS Criterion Model 42 electromechanical universal test system. T-peel tests were performed with a crosshead speed of 50 mm/min, according to the procedures in the ASTM D1876-08 standard. As shown in Fig. S1(a), nylon filter (0.45 μm pore) and polycarbonate filter (0.2 μm pore) are employed as the substrates for tested GPEs (dimension 30 mm × 10 mm × 1 mm), which prevent GPEs' elongation along the peeling direction. It can be observed in Fig. S1(b) that the soft PAZT GPE undergoes a cohesive failure near the interface during the T-peel test, leaving a residual layer on PHEC GPE. The failure within GPE indicates strong and durable adhesion at PAZT/PHEC interface, which is attributed to mutual solubility of GPEs due to having the same plasticizer (propylene carbonate). In Fig. S1(c), the T-peel test of PAZT/PHEC heterojunction gives an adhesion energy of $38.00 \pm 4.39 \text{ J/m}^2$ through 3 tested samples, comparable with the reported perfluorinated sulfonic acid (PFSA) GPEs [1]. Still, the cohesive failure occurs within the PAZT GPE instead of on the interface.

Fig. S1 Strong adhesion of PHEC GPE with PAZT GPE. (a) image of a T-peel test setup; (b) Image of cohesive failure during a T-peel test of a PHEC GPE adhered with a PAZT GPE; (c) the force-displacement curves for a PHEC GPE bonded with a PAZT GPE.

5. For the logic gates constructed using ionic diodes (both OR and AND), why the measured voltage under (0,0) state is not 0 but 0.4 V or -0.4 V?

The small voltage drifts at the (0,0) status are attributed to the charging and discharging-caused delays of the ionic diodes. In the ionic logic gates, because we tested the logic gates by using (1,0) first, the charge accumulated within (1,0) state has effects on the next (0,0) state. Thus, the voltage at (0,0) starts at a value of ~ 0.4 V for OR gate and ~ -0.3 V for AND gate, which would gradually decay toward 0 V. This phenomenon is also observable in the ionic logic gates constructed with hydrogel-based ionic diodes [1], whose measured voltage under (0,0) state is ~ -0.5 V (OR gate) and ~ 3.0 V (AND gate). The ionic OR gate built with multiwalled carbon nanotubes ionic diodes by Peng's research group also displayed a voltage of 0.69 V under (0,0) state [2].

Reviewer #2 (Remarks to the Author):

The authors have reported an ionic liquid based ionic diode by using two gel polymer electrolytes. They suggested the new mechanism of ionic diode by using the difference in ion solubility in two gels. By using the ionic liquid, they secured the thermal stability than ordinary hydrogel ionic diode. However, there are some unclear factors and major points that this reviewer would like to ask. Please consult the followings:

1. The authors said the ionic diode is bioinspired, but it is difficult to say that there are similarities between the neuron ion channel and the ionic diode as the working principle is completely different. The neuron ion channel uses ion selective membranes, but the ionic diode uses the ion solubility difference between two gel polymer electrolytes. In terms of functionality, the neuron ion channel only moves Na^+ ions in one direction,

whereas ionic diode moves anions in the opposite direction along with movement of cations. It is hard to say that gel polymer electrolyte ionic diode is bioinspired without further explanation.

Thank you for the suggestion. The title has been revised by removing the word “bioinspired”. And the part illustrating the mechanism has been rewritten:

As depicted in Fig.1(a), $[\text{EMIM}]^+$ and $\text{CF}_3\text{O}_3\text{S}^-$ ions are mobile and well-dispersed in their respective GPEs. Since Cl^- ions from PHEC exhibit limited ion diffusion in PAZT, the Cl^- ions experience restricted ion transport when passing through GPE interface and entering PAZT. Likewise, Zn^{2+} ions from PAZT are difficult to diffuse into PHEC for the same reason. Once the two GPEs are in contact, Zn^{2+} and Cl^- would diffuse and accumulate at the interface to form an IDL due to ion concentration gradient across the device, ceasing further diffusion of mobile $[\text{EMIM}]^+$ and $\text{CF}_3\text{O}_3\text{S}^-$ through the interface.

Fig. 1 Working mechanism of GPE ionic diode. (a) schematic of GPE ionic diode without voltage

bias; (b) under forward bias; (c) under reverse bias; (d) schematic of biological neuron ion channel.

Under a forward bias (Fig. 1(b)), $[\text{EMIM}]^+$ ions in the PHEC and $\text{CF}_3\text{O}_3\text{S}^-$ ions in the PAZT GPE would be drawn towards the interface (Fig. 1(b)). Owing to the high migration rates of these ions, they are less obstructed in the GPE and could pass through the interface into the other GPE. In other words, the forward voltage bias eliminates the IDL constructed by Zn^{2+} and Cl^- and assists the ions to flow through the device, leading to a relatively high current in the device and forward conduction. When a reverse bias is applied on the GPE ionic diode (Fig. 1(c)), the IDL would be enhanced by greater $\text{Zn}^{2+}/\text{Cl}^-$ interfacial accumulation, thereby, further restraining ion transport of $[\text{EMIM}]^+$ and $\text{CF}_3\text{O}_3\text{S}^-$. This thicker IDL behaves like a widened depletion region in the conventional p-n junction with a stronger field. The ion flow is inhibited by the interface, performing a reverse cut-off that could be observed in common Si-based diodes.

The working principle of the GPE ionic diode lies in the difference in ion diffusion/solubility in the GPE heterojunction, while biological ion channel (Fig. 1(d)) relies on chemical activation and specific ion recognition. When compared with ion channel, the GPE ionic diode shares certain similarities in their approach of controlling ion transport. The ion rectification of GPE ionic diode is realized via permitting or restricting the flowing of certain ions through the GPE heterojunction, comparable in certain respects of ion-selective permeability in biological ion channels. In addition, unlike typical ion rectification devices which have fixed charges on polymer chains or surfaces, GPE ionic diode also shares common feature with biological system, that is, they both allow free ion transport of positive and negative ions.

2. In figure 3 & figure 4, there are many experiments about performance of the ionic diode. However, experiments about durability of the ionic diode are missing. Additional experiments are required to confirm the durability of the ionic diode (Cycle test, maximum operating voltage range test, maximum operating time test, etc.).

To test the stability of ion rectification in GPE ionic diode, a prolonged period of ± 1.0 V is applied. As shown in Fig. S3(a), the current-time curves produced by the ± 1.0 V remains to be smooth and stable in the 200 s measurement, indicating good durability for long-term operation. The rectifying ratio (η) reaches high value of 24.0 at 3.5 s and starts to decrease at 12.6 s. The rectifying ratio decays to 80% ($\eta = 19.2$) at 29.5 s and 50% ($\eta = 12.0$) at 114.4 s, finally 41.4% ($\eta = 9.95$) at the 200 s. Moreover, the durability of the GPE ionic diode under a square-wave voltage of ± 1 V with 0.1 Hz frequency was also measured and displayed in Fig. S3(b). The rectifying ratio of GPE ionic diode decreased to 75.8% of initial performance after 50 cycles, then 56.2 % of initial performance remains after 100 cycles of alternating voltage.

Fig. S3 (a) Transient current response of PAZT/PHEC heterojunction at ± 1.0 V for 200 s; (b) performance stability of the PAZT/PHEC diode a square-wave voltage of ± 1.0 V with 0.1 Hz frequency.

The maximum operating voltage of the GPE ionic diode is from -2 V to $+2$ V. If the CV scan is stretched to a wider potential range from -3 V to $+3$ V (Fig. R2), faradaic (or redox) reaction comes into play, to which a prominent cathodic peak at ~ -0.7 V appears alongside a large current density reaching 0.8 mA/cm^2 during the anodic scan. This phenomenon could be due to electrolyte electrolysis and possible zinc metal deposition [3]. Thus, the working range shall be restrained within ± 2 V to ensure that the GPE ionic diode is working via the ion-transport-based mechanism rather than a faradaic process.

Fig. R2 CV plots for PAZT/PHEC scanned from -3 V to $+3$ V.

3. Mechanism of the ionic diode is only possible with certain combinations of ions. Are there any other ions & gels combinations that the device can work with?

Yes, other ion combinations with ion rectifying ability have also been explored. Since the ion rectification is based on the difference in ion solubility in the GPE heterojunction, this mechanism is versatile when similar conditions are met. In our early trials as shown in Fig. R3(a), PMMA:[BMIM]BF₄/ PVDF:Li[TFSI] heterojunction was tested at ± 1.0 V but only produces a negligible rectifying ratio of 1.16 due to the higher mobility of the Li ions as compared with the Zn ions that we use in this work. The non-rectifying result is in line with the solubility tests we have summarized in Table S4. Another successful ion combination that we conducted is PMMA:CH₃COONH₄ (PACN)/ PVDF-HFP:[EMIM]Cl GPE heterojunction. As shown in Fig. R3(b), this heterojunction produces a rectifying ratio of 5.40 under ± 1.0 V. In addition, Q-V plots with symmetry and asymmetry could be observed in the PACN homojunction and PAZN/PHEC heterojunction, respectively (Fig. R3(c)).

Fig. R3 (a) Transient current response of PMMA:[BMIM]BF₄/ PVDF:Li[TFSI] heterojunction at ± 1.0 V; (b) PACN/PHEC heterojunction at ± 1.0 V; (c) Q-V plots of PACN/PHEC GPE heterojunction and homojunctions from - 1 V to + 1 V.

Reviewer #3 (Remarks to the Author):

I looked through the article and suggest major revisions based on the comments below.

The novelty of this work is an ionic diode that is not based on water or polyelectrolytes and can work at High temperatures.

The authors need to make a better comparison between different ionic diode systems in this work to show where this system is better in terms of quantitative metrics and possibly also areas of use. The comparison between this Ionic diode and the bio channels is in my opinion not very instructive and indeed even misleading: After all

this system is not water based whereas biology is and so are already reported ion rectifiers. Second, the ion channels act based on a completely different mechanism where chemical energy and specific ionic recognition shuffle ions.

I suggest to remove the word bioinspired from the title and also rewrite the parts that make the direct comparison between this system and biological ion channels.

Thank you for your suggestion. The title has been revised by removing the word “bioinspired”. And the part illustrating the mechanism has been rewritten:

As depicted in Fig.1(a), $[\text{EMIM}]^+$ and $\text{CF}_3\text{O}_3\text{S}^-$ ions are mobile and well-dispersed in their respective GPEs. Since Cl^- ions from PHEC exhibit limited ion diffusion in PAZT, the Cl^- ions experience restricted ion transport when passing through GPE interface and entering PAZT. Likewise, Zn^{2+} ions from PAZT are difficult to diffuse into PHEC for the same reason. Once the two GPEs are in contact, Zn^{2+} and Cl^- would diffuse and accumulate at the interface to form an IDL due to ion concentration gradient across the device, ceasing further diffusion of mobile $[\text{EMIM}]^+$ and $\text{CF}_3\text{O}_3\text{S}^-$ through the interface.

Fig. 1 Working mechanism of GPE ionic diode. (a) schematic of GPE ionic diode without voltage bias; (b) under forward bias; (c) under reverse bias; (d) schematic of biological neuron ion channel.

Under a forward bias (Fig. 1(b)), $[EMIM]^+$ ions in the PHEC and $CF_3O_3S^-$ ions in the PAZT GPE would be drawn towards the interface (Fig. 1(b)). Owing to the high migration rates of these ions, they are less obstructed in the GPE and could pass through the interface into the other GPE. In other words, the forward voltage bias eliminates the IDL constructed by Zn^{2+} and Cl^- and assists the ions to flow through the device, leading to a relatively high current in the device and forward conduction. When a reverse bias is applied on the GPE ionic diode (Fig. 1(c)), the IDL would be enhanced by greater Zn^{2+}/Cl^- interfacial accumulation, thereby, further restraining ion transport of $[EMIM]^+$ and $CF_3O_3S^-$. This thicker IDL behaves like a widened depletion region in the conventional p-n junction with a stronger field. The ion flow is inhibited by the interface, performing a reverse cut-off that could be observed in common Si-based diodes.

The working principle of the GPE ionic diode lies in the difference in ion diffusion/solubility in the GPE heterojunction, while biological ion channel (Fig. 1(d)) relies on chemical activation and specific ion recognition. When compared with ion channel, the GPE ionic diode shares certain similarities in their approach of controlling ion transport. The ion rectification of GPE ionic diode is realized via permitting or restricting the flowing of certain ions through the GPE heterojunction, comparable in certain respects of ion-selective permeability in biological ion channels. In addition, unlike typical ion rectification devices which have fixed charges on polymer chains or surfaces, GPE ionic diode also shares common feature with biological system, that is, they both allow free ion transport of positive and negative ions.

There are many grammatical errors for example biological system relies (should be rely), is selected for their (should be its), are difficult to travel (should have lower diffusion rate or similar). This language must be reworked throughout.

Thanks for the suggestion. We have thoroughly re-worked on the writing to revise the grammatical errors.

Another problem is that the application of Triboelectric generator simply does not make sense to do with this system. The generator is purely electrical, and a regular solid state diode has much better properties for this application. The authors say that this diode is suitable for neuromorphic devices in conclusion but have no application or data to verify this statement and ideally also show a simple application where they use the device for analog Computation. In my opinion, if this article is to be published here there should at minimum be an application that is clearly relevant for this particular device, for example something which relies on high temperature above 100°C where this device is superior, and/or a system where the input to the diode is itself based on iontronic systems (eg an ionic sensor connected to diodes for computation or something like that)

Due to the intrinsic deformability endowed by the polymeric materials, our iontronic system could offer flexibility and stretchability which is attractive for applications in futuristic wearable devices and in-vivo implants when integrated with TENG. With that, our work illustrates a potential alternative to conventional rigid solid-state diodes. We have further verified the suitability of the GPE ionic diode for neuromorphic device and is included in our manuscript as follows:

In order to explore the potential of GPE ionic diode in synaptic devices (neuromorphic properties), a sequence of short voltage pulses was applied and the corresponding responses of GPE ionic diode were analyzed. Fig. 6(a) shows a typical excitatory postsynaptic current (EPSC) induced by a presynaptic voltage pulse of -1 V with a duration of 140 ms. The current increases with the voltage spike and reaches the peak at 4.79 μA , then decays gradually to the baseline. As shown in Fig. 6(b), the produced EPSC increases when the duration of presynaptic voltage (or the pulse width) is extended. The EPSC starts from 2.19 ± 0.16 μA at a pulse width of 70 ms and is saturated at a pulse width of 1400 ms, reaching a maximum of 12.47 ± 0.35 μA . This phenomenon is actuated by ion redistribution when the voltage pulse breaks the equilibrium in the GPE heterojunction, which resembles an EPSC in a biological synapse. The responses to incremental pulse amplitude from 0.1 V to 2.0 V with a constant pulse width of 140 ms are also investigated and shown in Fig. 6(c). Due to the demand for intelligent applications in aerospace, deep-well drilling, and high-speed automobiles [36], EPSC is also demonstrated at an elevated temperature (Fig. 6(d)). Under a high temperature of 100 $^{\circ}\text{C}$, the EPSC starts from 3.35 ± 0.02 μA at 70 ms pulse and plateaus at 20.07 ± 0.15 μA at 700 ms pulse. The premature saturation and higher EPSC is attributed to the accelerated ion diffusion and migration under high temperature environment.

Fig. 6 (a) EPSC induced by a -1 V presynaptic voltage with the duration of 140 ms; (b) EPSC as a function of spike duration induced by the -1 V presynaptic voltage; (c) EPSC as a function of presynaptic voltage with spike width maintained at 140 ms; (d) EPSC as a function of spike duration induced by the -1 V presynaptic voltage under high temperature of 100 °C.

References

- [1] Wang, Y., Wang, Z., Su, Z., Cai, S. Stretchable and transparent ionic diode and logic gates. *Extreme Mechanics Letters* 28, 81-86 (2019).
- [2] Peng, R., Pan, Y., Li, Z., Zhang, S., Wheeler, A. R., Tang, X., Liu, X. Ionotronics based on horizontally aligned carbon nanotubes. *Advanced Functional Materials* 30, 2003177 (2020).
- [3] Xue, Y., Zhou, Z. P., Yan, Y. D., Zhang, M. L., Li, X., Ji, D. B., Tang, H., Zhang, Z. J. Electrochemistry of Zn and co-reduction of Zn and Sm from LiCl-KCl melt. *RSC Advances* 5, 23114-21 (2015).

REVIEWERS' COMMENTS

Reviewer #1 (Remarks to the Author):

The reviewer has studied the rebuttal letter and revised the manuscript in detail. The revisions are satisfactory and can be accepted without changes.

Reviewer #2 (Remarks to the Author):

The authors properly addressed the comments from this reviewer. I recommend publication of this work.

Reviewer #3 (Remarks to the Author):

This revision has addressed all the questions I raised.